# Prevalence of Methicillin and β−Lactamase Resistant Pathogens Associated with Oral and Periodontal Disease of Children in Mymensingh, Bangladesh

**DOI:** 10.3390/pathogens11080890

**Published:** 2022-08-09

**Authors:** Sharmin Sultana, Rokshana Parvin, Mst. Sonia Parvin, Md. Taohidul Islam, Abu Saleh Mahfuzul Bari, Emdadul Haque Chowdhury

**Affiliations:** 1Department of Pathology, Faculty of Veterinary Science, Bangladesh Agricultural University, Mymensingh 2202, Bangladesh; 2Population Medicine and AMR Laboratory, Department of Medicine, Faculty of Veterinary Science, Bangladesh Agricultural University, Mymensingh 2202, Bangladesh

**Keywords:** OPD pathogens, virulence genes, β−lactamase resistant, methicillin resistant, pediatric patient

## Abstract

Oral and periodontal diseases (OPD) is considered one of the main problems of dentistry worldwide. This study aimed to estimate the prevalence of oral and periodontal pathogenic bacteria along with their antimicrobial resistance pattern in 131 children patients aged between 4–10 years who attended in Mymensingh Medical College Hospital during October 2019 to March 2020. OPD pathogens were identified through isolation, cultural and biochemical properties, and nucleic acid detection. The isolates were subjected to antimicrobial susceptibility to 12 antibiotics commonly used in dentistry. In addition, the isolates were analyzed molecularly for the presence of six virulence and three antibacterial resistance genes. Five pathogens were identified, of which *Staphylococcus aureus* (*S. aureus*) (49%) and *S. salivarius* (46%) were noticed frequently; other bacteria included *S. mutans* (16.8%), *S. sobrinus* (0.8%) and *L. fermentum* (13.7%). The virulence genes—clumping factor A (*clfA*) was detected in 62.5% isolates of *S. aureus*, and gelatinase enzyme E (*gelE*) gene was detected in 5% isolates of *S. salivarius,* while other virulence genes were not detected. All the tested isolates were multidrug-resistant. The overall prevalence of MDR *S. aureus*, *Streptococcus* spp. and *L. fermentum* was 92.2%, 95.1% and 100%, respectively. It was observed that a high proportion of isolates were found resistant to 5–8 antibiotics. A majority of *S. aureus*, *Streptococcus* spp., and *L. fermentum* isolates tested positive for the β−lactamase resistance genes *blaTEM* and *cfxA*, as well as the methicillin resistance gene *mecA*. Phylogenetically, the resistance genes showed variable genetic character among Bangladeshi bacterial pathogens. In conclusion, *S. aureus* and *S. salivarius* were major OPD pathogens in patients attended in Mymensingh Medical College Hospital of Bangladesh, and most were Beta-lactam and methicillin resistant.

## 1. Introduction

Oral and periodontal disease (OPD) is one of the most prevalent diseases in the world, causing significant morbidity, particularly among children [1]. The human oral flora encompasses over 700 microorganisms, 50% of them being uncultivable microbes [2]. Bacteria, fungi, protozoa and other ecological community of biofilm cling to various surfaces of the mouth cavity to make up the oral flora [3,4]. Some of these microorganisms, mostly bacterial species, are associated with human diseases in certain conditions [5,6,7]. Various species of the genus *Streptococcus*, *Staphylococcus*, *Lactobacillus*, *Enterococcus*, *Corynebacterium*, *Veillonella* and *Bacteroids* are predominant bacteria in the human oral cavity. Mutans group *Streptococci* (*Streptococcus mutans* and *Streptococcus sobrinus*), *mittis* group *Streptococci* (*Streptococcus sanguinis*, *Streptococcus oralis*, and *Streptococcus gordonii*) and *Salivarius* group (*Streptococcus salivarius*) are major constituents of human oral flora [8,9,10,11]. High levels of *S. mutans* and early colonization are considered key risk factors for the development of dental caries in both children and adults [12,13]. Historically, *Lactobacilli* were the first microorganisms implicated in dental caries development [14]. They appear in large quantities in saliva, on the dorsum of the tongue, mucous membranes, the hard palate, dental plaque and, in smaller numbers, on tooth surfaces throughout a child’s earliest years of life [15]. Aside from microflora, other risk factors, particularly obesity and tooth caring, are linked to microbially associated inflammatory disease affecting tooth-supporting structures [16,17].

*S. aureus* is a well-known human pathogen that causes bacteriaemia, pneumonia, osteomyelitis, acute endocarditis, myocarditis, pericarditis, encephalitis, meningitis, choriomeningitis, mastitis, scalded skin syndrome and abscesses of the muscle, urogenital tract and various intra-abdominal organs [18]. Above all the diseases, osteomyelitis is the most common disease in the oral surface area. Methicillin-resistant *S. aureus* (MRSA) has spread widely to become a major clinical and epidemiological problem in many medical centers [19,20,21,22]. Antibiotics are the only treatment option against bacterial infection. Beta-lactam, tetracycline and macrolide antibiotics have been prescribed in endodontics, especially for the treatment of acute apical extension or systemic involvement, spreading infection and abscess in medically compromised patients who are at increased risk of a non-oral secondary infection after bacteremia, prophylaxis for medically compromised patients during routine endodontic therapy and replantation of avulsed teeth [23]. Antibiotic prophylaxis before a dental procedure reduces the frequency, nature or duration of bacteremia. The American Dental Association (ADA), The American Heart Association (AHA) and the American Academy of Orthopedic Surgeons (AAOS) recommend cephalexin, cephradine or amoxicillin for prophylactic use in dental procedures [24]. Regular and frequent use of antibiotics in dental infection often causes public health troubles by leading to the development of resistant microbes, including multidrug-resistant pathogens [25]. According to WHO, antimicrobial resistance (AMR) is an emerging problem worldwide and one of the most important threats to human, animal and environmental health [26]. However, there is a paucity of information on the burden of OPD pathogens, including the antimicrobial resistance pattern in children in Bangladesh. In the present study, OPD bacteria were isolated and detected, and virulence factors and AMR patterns were determined among children admitted in Mymensingh Medical College Hospital of Bangladesh.

## 2. Results

### 2.1. Isolation and Morphologic Identification of the Pathogen

Bacteria were isolated and morphologically identified as specific colony features were observed in the selective agar media and Gram staining characteristics (Figure 1).

Further formation of bubble at biochemical test indicates catalase positive, fermentation of basic sugar and formation of gas indicates positive sugar fermentation test, no color changes indicate negative results of MR, Indole and VP test for the bacteria. Altogether, *S. aureus* (n = 64), *S. salivarius* (n = 60), S. mutans (n = 22), *S. sobrinus* (n = 1) and *L. fermentum* (n = 18) were the five OPD pathogens detected (Table 1). *S. aureus* was the highest (48.9%) among 131 isolates; multiple co-infections were also detected. Of the 131 children, 100 displayed OPD clinical symptoms, whereas the remaining 31 were found asymptomatic. Bacteria were isolated successfully among asymptomatic children.

### 2.2. Antimicrobial Resistance Pattern of the Organisms

The analysis revealed that more than 90% of *S. aureus* isolates showed resistance to five commonly used antibiotics. Within MDR *S. aureus*, the highest resistance was detected against metronidazole (100%, n = 64) followed by cefixime (96.9%, n = 62), cephradine (95.3%, n = 61), cefuroxime (92.2%, n = 59) and amoxicillin (90.6%, n = 58) (Table 2). The lowest resistance was observed against ciprofloxacin (4.7%, n = 3), tetracycline (12.5%, n = 8) and moxifloxacin (23.4%, n = 15).

In the case of *Streptococcus* spp., the highest resistance was detected against metronidazole (98.8%, n = 82) followed by cephradine (97.6%, n = 81) and amoxicillin (91.6%, n = 76). The lowest resistance was observed against ciprofloxacin (4.8%, n = 4), moxifloxacin (4.8%, n = 4) and chloramphenicol (9.6%, n = 8) (Table 1). On the other hand, *L. fermentum* revealed the highest resistance to cefixime (100%, n = 18), moxifloxacin (100%, n = 18), cefuroxime (100%, n = 18) and amoxicillin (100%, n = 18), followed by metronidazole (94.4%, n = 17) and cephradine (88.9%, n = 16). The lowest resistance was observed against chloramphenicol (5.6%, n = 1) and azithromycin (16.7%, n = 3) (Table 2).

The prevalence of MDR *S. aureus*, *Streptococcus* spp. and *L. fermentum* was 92.2%, 95.1% and 100%, respectively (Figure 2). Among these three bacteria, 57 to 63% of the isolates were resistant to 3–4 antimicrobial classes, 25 to 30% isolates to 5−6 and 4 to 11% of isolates to ≥ 7 classes of antimicrobials (Figure 2). Among 64 *S. aureus*, 83 *Streptococcus* spp. and 18 *L. fermentum* isolates, all the isolates were resistant to at least one and up to 12 antimicrobial agents tested.

### 2.3. Frequency of Virulence Genes in OPD Pathogens

The results showed that two virulence genes—clumping factor A (*clfA*) for *S. aureus* (n = 40) and Gelatinase enzyme E (*gelE*) for *S. salivarius* (n = 3) were detected. The other four virulence genes such as collagen-binding protein (*ace*), endocarditis antigen (*efa*), hyaluronidase enzyme (*hyl*), cytolysin activator (*cylA*) were not detected in any of the tested isolates.

### 2.4. Frequency of Resistance Genes Associated with OPD in Children

Three isolates of *S. aureus* (4.7%), 10 isolates of *Streptococcus* spp. (12.1%) and 11 isolates of *L. fermentum* (61.1%) were positive for the *blaTEM* gene (Table 3). The *CfxA* gene was detected in 60 isolates of *S. aureus* (93.8%), 79 isolates of *Streptococcus* spp. (95.2%) and 13 isolates of *L. fermentum* (72.2%). The *mecA* gene was used to confirm the presence of *S. aureus* in 8 isolates out of 64 (12.5%). There was no *nim* gene detected in tested isolates of *S. aureus*, *Streptococcus* spp. and *L. fermentum*.

### 2.5. Phylogenetic Analyses of Selected Species Identifying, Virulence and AMR Genes

A total of 20 nucleotide sequences of the *nuc* gene of *S. aureus* were subjected to analyses. From the phylogeny, Bangladeshi isolates shared genetic similarities with worldwide isolates, with six of them being the most closely linked to the oldest USA–2006 strain. {reference strain of NCTC−8325 (GenBank: NC_007795.1)}. Three Bangladeshi isolates were closely related to China−2007 and Denmark–2018 (Figure 3A). Within the Bangladeshi isolates, however, there was significant genetic variety, with a single Bangladeshi isolate (BAU-OS52Sa) maintaining a long distance from the others. Similarly, 18 nucleotide sequences from the *gtfK* gene of *S. salivarius* were shown to be genetically comparable to global isolates, with five isolates being the most closely linked to the oldest Australian isolate (accession no. Z11872.1) from 1992 (Figure 3B). Four Bangladeshi *S. salivarius* were closely related to China–2016 and two were found to be firmly connected to China–2017 and Iran–2014. The sequences of *If* gene of *L. fermentum* revealed five Bangladeshi isolates genetically closely related to United Kingdom–1993, Nigeria–2020 and Australia–2020 (Figure 3C). Five other isolates were closely related to USA–2011.

A total of 21 nucleotide sequences of virulence gene, clumping factor A (*clfA*)from *S. aureus* was subjected to analysis. The analysis revealed a single isolate closely related to an oldest Australia–1995 isolate, another single to an isolate from Switzerland-2000 (Figure 3D). The sequence analyses of the β-lactamase-resistant *blaTEM* (Figure 3E) and methicillin-resistant *mecA* (Figure 3F) gene of their respective bacteria, on the other hand, found them to be genetically similar with some global isolates that showed antimicrobial resistance as well. Overall, respective genes of Bangladeshi isolates showed genetic variation among themselves.

## 3. Discussion

This study described the isolation, identification and AMR of pathogenic bacteria from OPD pediatric patients in Mymensingh, Bangladesh. *Staphylococcus aureus*, *Streptococci* species and *Lactobacilli* species have been strongly associated with dental caries [27,28,29,30]. Our results also confirmed the presence of *S. aureus* (48.9%), *S. salivarius* (45.8%), *S. mutans* (16.8%), *S. sobrinus* (0.8%) and *L. fermentum* (13.7%). Catalase, coagulase and thermo-nuclease enzymes are important phenotypic markers of *S. aureus.* In addition, PCR analysis using 16S rRNA primers is required for the confirmation of species [18]. Furthermore, the *nuc* gene is used for identification for *S. aureus* strains [31,32,33]. Similarly, in this study, *L. fermentum*, *S. mutans* and *S. sobrinus* were the most frequent isolates that were found in this study and also are well known inhabitants of the human oral microflora [34]. It is recognized that the streptococci of the mutans group play an important role, as the major cariogenic agents in the etiology of dental caries. It is important to recognize the distribution of *S. mutans* and *S. sobrinus*, which are the most commonly found species in humans. Their correct identification and differentiation from other species is considered an important step in understanding the early phases of bacterial colonization of dental plaque [35,36]. Although, several other studies indicated the most frequently isolated species are *S. mittis*, *S. mutans* and *S. salivarius* from the oral cavity of patients worldwide [37,38,39]. The difference in prevalence of OPD pathogens could be due to different global food habits, hygienic practices and environments.

Odontogenic infections can range in severity from mild to moderate to severe to emergency, depending on factors such as the virulence of the microorganism involved, the amount of pathogen within tissues, the anatomy of the afflicted area and the patients’ general health [40]. Genes such as clumping factor A (*clfA*), which encodes bacterial surface components that recognize sticky matrix molecules, and gelatinases enzyme (*gelE*), which degrades gelatin, are significant virulence genes in biofilm production [41,42]. The *clfA* and *gelE* genes were discovered in 40 (62.5%) and 3 (5%) of the strains in this study, respectively, and have been found in many previous studies at varied percentages [43,44].

Dental patients usually take antibiotics primarily to treat postoperative and secondary infections. β−lactams are one of the most frequently prescribed classes of antibiotics for both medical and dental purposes. However, there is substantial concern about the compromised efficacy of these antimicrobials due to the development of bacterial resistance [45]. Antibiotics are often ineffective in preventing alveolitis after erupted tooth extractions [46]. Most of the resistance is due to one of the plasmid or transposons encoded β−lactamases such as *blaTEM*, *cfxA*. These genes typically remain active against ampicillin, amoxicillin, penicillin, and, to a lesser extent, third generation antibiotics [47]. In this study, >80% of all the five microorganisms *S. aureus*, *Streptococcus* spp. and *Lactobacillus* spp. were found resistant against metronidazole, β−lactam antibiotics (cephradine, amoxicillin), cefixime and cefuroxime, which were most commonly used in dentistry in Bangladesh. In addition, >50% of these organisms were sensitive to ciprofloxacin, tetracycline, chloramphenicol and moxifloxacin. Similar antimicrobial resistance and susceptibility results were reported by other authors [23,48,49,50]. The excess and unplanned use of amoxicillin significantly increases the chance of production of resistance bacteria [51]. Previous studies [52,53,54], suggested that subgingival pathogens are resistant to β−lactam antibiotics. J. Van Winkelhoff et al. (1997) found a correlation between the observed frequency of the *blaTEM* gene and the high intake of the β−lactam class of antibiotics [52,55,56,57]. According to the European Centre for Disease Prevention and Control report 2013, Greece had one of the highest rates of antibiotic consumption among the European countries, and β−lactams were among the most frequently prescribed antibiotics [58]. Jungermann et al. (2011) found that *blaTEM* was the most prevalent antibiotic resistance gene in samples from primary and secondary root canal infections [59]. About 93.8% of *S. aureus*, 95.2% of *Streptococcus* spp. and 72.2% *L. fermentum* isolates were found resistant to *cfxA* in the present study and about 4.7% of *S. aureus*, 12% of *Streptococcus* spp. and 61% *L. fermentum* isolates were found resistant to *blaTEM* in the present study.

Staphylococci are known to be frequent colonizers of the oral cavity, and the incidence of methicillin resistance in oral staphylococci is poorly studied [60,61,62]. The identification of methicillin-resistant staphylococci (MRSA) in the laboratory is sometimes complicated by the heterogenous expression of resistance and the variables that influence this expression (i.e., pH, temperature and salt concentrations). The present study showed that 12.5% (8/64) methicillin-resistant (*mecA* gene) staphylococci were found in the oral cavity of children as found by Das et al. in 2019 [63]. It is a new gene that is emerging in the world, especially in European countries, and spreading among human and animals throughout the world, which follows the statement of Paterson et al. (2014) [64] but not for Bangladesh. Phylogenetically, the resistance genes also showed variable genetic character among Bangladeshi bacterial pathogens as well as with global isolates.

Many risks factors for colonization of pathogenic bacteria in OPD pediatric patients have been analyzed [29]. The current study is the evidence of the presence of symptomatic or asymptomatic carriers of pathogenic bacteria. Dental patients are not the only group responsible for spreading virulence or antimicrobial resistance genes containing the pathogen; health professionals may transmit this pathogen through their instruments as well [65]. Since most of the dental caries associated pathogens are multidrug resistant, dentists should have a well-established protocol for regulating the spread of such a pathogen. The U.S. Centers for Disease Control and Prevention (CDC) does, however, propose some routine preventative measures that should be followed in Bangladesh.

## 4. Materials and Methods

### 4.1. Selection of Children and Sample Collection

Between October 2019 and March 2020, 131 children aged 4 to 10 years old who were registered at the Outdoor Dental Unit of Mymensingh Medical College Hospital (MMCH) with oral and periodontal disease (OPD) were chosen for this study. In OPD, the children’s complaints were eruption pain, loose teeth, swelling and pus discharges from teeth and gums, caries, gingivitis, aphthous ulcers, and broken and sharpened teeth. The researcher, a registered dental surgeon, examined the children under the head lamp illumination with a disposable dental mirror and a ball-ended Community Periodontal Index (CPI) probe. Children showing clinical signs along with asymptomatic children were also included. Before taking the consent, objectives of the study along with its procedure, risk and benefit were explained to the parents or legal guardians in easily understandable local language. However, children with any systemic diseases such as kidney, heart and liver diseases were excluded. Furthermore, children with any acute infections, fever, diarrhea during clinical examination and difficulty in opening the mouth were also excluded. Oral swab samples were collected by sterile swab stick inverted in phosphate buffer saline (PBS).

### 4.2. Ethical Consideration

Ethical clearance was obtained from the Ethical Standard of Research Committee, Bangladesh Agricultural University Research System (BAURES/ESRC/VET/15-1). In addition, a written permission was taken from the Head of the Dental department and Director of Mymensingh Medical College Hospital, Mymensingh.

### 4.3. Culture Techniques

All samples were managed by code number to conserve children’s confidentiality. The samples were initially inoculated in nutrient broth and incubated overnight at 37 °C temperature. Then each broth culture was streaked onto Mittis Salivarius agar (HiMedia, Maharashtra, India), Mannitol Salt agar (HiMedia, Maharashtra, India), and De Man Rogosa Sharpe or Lactobacillus agar (HiMedia, India) for isolation of the target pathogen *Streptococcus* sp., *S. aureus* and *L. fermentum*, respectively. All presumptive colonies were tested for biochemical properties (catalase, sugar fermentation test, methyl red test, Voges–Proskauer test, indole test) and Gram staining was performed following previous protocol to identification of the bacteria [66].

### 4.4. Antimicrobial Susceptibility Testing (AST)

The antimicrobial resistance (AMR) profile of all isolates was determined using the Kirby–Bauer disk diffusion method as described by the Clinical and Laboratory Standards Institute [67]. A panel of 12 antibiotics representing 8 different antimicrobial classes such as clindamycin (CD, 2 µg); azithromycin (AZM, 15 µg); erythromycin (E, 15 µg); amoxicillin (AMX, 30 µg); cefradine (CH, 25 µg); cefuroxime (CXM, 30 µg); cefixime (CFM, 5µg); moxifloxacin (MO, 5µg); ciprofloxacin (CIP, 5 µg); tetracycline (TE, 30 µg); chloramphenicol (C, 30 µg); and metronidazole (MT, 5 µg) were used for *S. aureus*, *Streptococcus* sp. and Lactobacillus. For performing disk diffusion method, at first pure cultured colony was diluted in sterile nutrient broth (NB) and adjusted to opacity equivalent to a 0.5 McFarland turbidity standard and suspensions were spread on the surface of Mueller-Hinton agar (MHA, Himedia, Maharashtra, India). Finally, the antimicrobial disks were placed and incubated at 37 °C for 18 h. The interpretive category (susceptible, intermediate and resistant) of each isolate was determined as per the CSLI guidelines [68]. Furthermore, any isolate that showed resistance to at least one antimicrobial agent in three or more antimicrobial classes was defined as multidrug resistant (MDR) [69].

### 4.5. Molecular Identification of the Pathogen Specific, Virulence and Resistance Genes

The pure isolates of the organisms were sub-cultured overnight in NB and genomic DNA was extracted by using the “boiling” method as described [70]. Briefly, 1 mL of aliquot of overnight cultures was taken in 1.5 mL DNase/RNase free Eppendorf tube using DNase/RNase free pipette tips and centrifuged at 14,000× *g* rpm for 5 min at 40 °C. The supernatant was discarded and pellet was collected. The pellet was washed three times using phosphate buffered saline and re-suspended in 200 µL of DDW. The bacterial suspension was vortexed before each centrifugation until dissolution of the pellets. The suspension was boiled at 100 °C for 15 min. Just after boiling, the Eppendorf tube was placed in ice for 15 min and then centrifuged at 14,000× *g* rpm for 5 min. An aliquot of 100 µL of the supernatant containing genomic DNA was collected in an Eppendorf tube, and stored at −20 °C for further study.

Species identification was carried out through amplification of *nuc* gene of *S. aureus*, *gtfK* gene of *Streptococcus salivarius* (*S. salivarius*), *gtfb* gene of *Streptococcus* mutans (*S. mutants*), *gtfi* gene of *Streptococcus sobrinous* (*S. sobrinous*) and *lf* gene of *L. fermentum*. In addition, six virulence genes that were collagen-binding protein (ace), endocarditis antigen (*efa*), hyaluronidase enzyme (*hyl*), cytolysin activator (*cylA*), gelatinase enzyme (*gelE*) and clumping factor A (*clfA*) gene of the isolates were targeted. The presence of β- lactamase (*blaTEM* and *cfxA*) and methicillin resistance (*mecA*) genes from *S. aureus* and metronidazole resistance (*nim*) gene were determined by PCR. Target genes and primers are listed in Appendix A.

The target genes were amplified using 25 µL of final volume that included 12.5 µL master mix (New England Biolabs, Ipswich, UK); forward and reverse primers (10 pmol/µL) (2 µL); nuclease free water (5.5 µL) and DNA template (5 µL). PCR controls consisted of genomic DNA that were positive for relevant genes. Nuclease-free water was used as negative controls. The amplified PCR product was then visualized in 1.5% to 2% agarose gels (depending on band size) with ethidium bromide under UV transilluminator. The expected positive band was identified and compared using TriDye appropriate DNA ladder size (New England Biolabs, Ipswich, UK).

### 4.6. Phylogenetic Analysis

Finally, few selected species identifying (*nuc, gtfK, lf*), virulence (*clfA*), and resistance (*blaTEM, mecA*) genes were targeted for sequencing from commercial sources (Macrogen, Seoul, South Korea). A total of 56 PCR products were sequenced partially, of which 31 were causative OPD organisms, 10 were virulence genes and 15 were antimicrobial resistance genes. Different other representative gene sequences of isolates from different countries available in the GenBank were downloaded from NCBI GenBank. The obtained sequences were primarily run for blast analysis through NCBI Blast platform (https://blast.ncbi.nlm.nih.gov/Blast.cgi, accessed on 12 April 2022). The sequences were edited, aligned, and phylogenetic trees were established with the help of software such as BioEdit (https://bioedit.software.informer.com/7.2/, accessed on 15 June 2022) and Mega X [71]. The sequences were submitted in public platform GenBank and are available under the accession number mentioned in Appendix A.

## 5. Conclusions

Five major OPD pathogens were detected in the oral and periodontal surfaces of children. The majority of Bangladeshi OPD bacterial isolates have genetic diversity between them. In most strains of *S. aureus* and *S. salivarius*, the virulent *clfA* and *gelE* genes were found. Multidrug-resistant microorganisms account for more than 90% of OPD pathogens. The antibiotic resistance genes *blaTEM* and *cfxA* were found to be the most common in OPD isolates. As a result, increasing awareness of underlying causes of AMR in OPD pathogens is critical, with an emphasis on prevention and the proper therapeutic approaches by dentists. Further study is needed to find the relationship between antimicrobial phenotype and genotypes, as well as the distribution of resistant genes.

## Figures and Tables

**Figure 1 pathogens-11-00890-f001:**
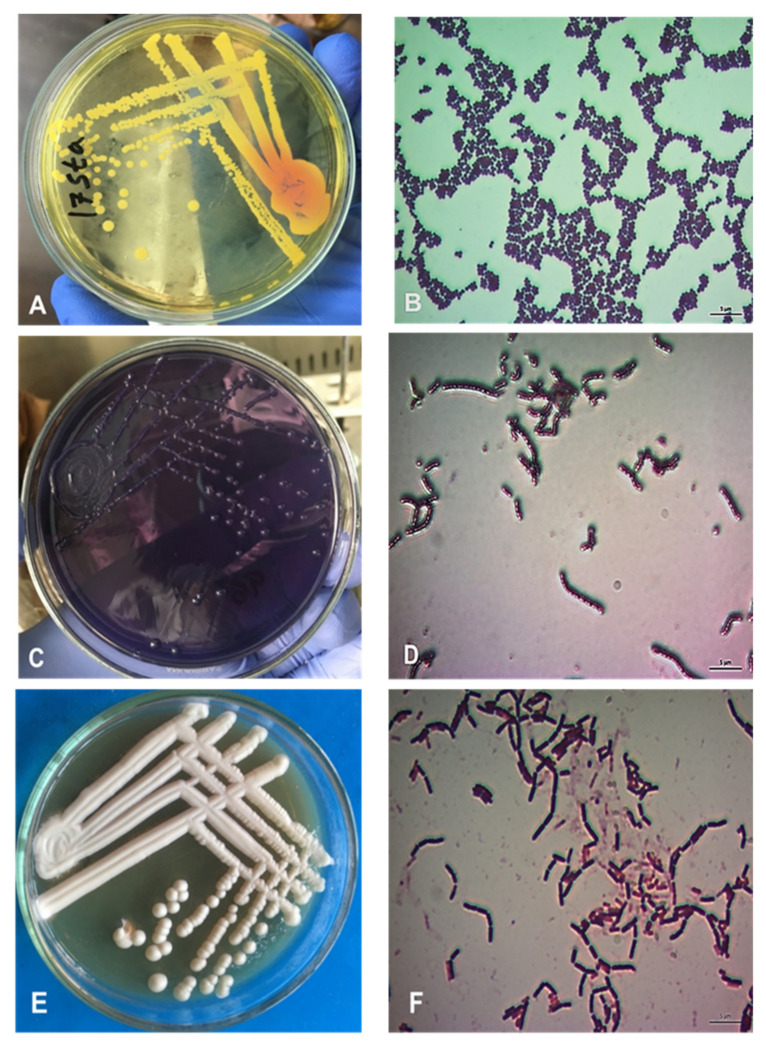
Morphology of *S. aureus*, *Streptococcus* and *Lactobacillus.* (**A**) Cultural properties showing yellowish color colony on Mannitol salt agar. (**B**) Gram staining of *S. aureus* showing pure culture gram positive cocci, arranged in grape like clusters. (**C**) Gum bubble colonies of *Streptococcus* spp. on Mittis Salivarius agar. (**D**) Gram-positive cocci showed chainlike clusters under light microscope. (**E**) White colonies on De Man Rogosa Sharpe agar. (**F**) Rod shape Gram-positive *Lactobacillus* under light microscope (100×).

**Figure 2 pathogens-11-00890-f002:**
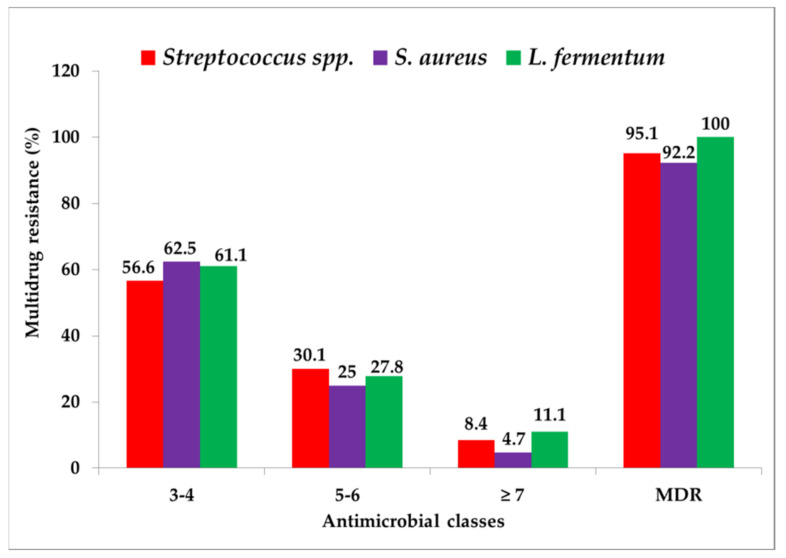
Multidrug resistance patterns observed among *Streptococcus* spp., *S. aureus* and *L. fermentum* isolated from children with oral and periodontal diseases.

**Figure 3 pathogens-11-00890-f003:**
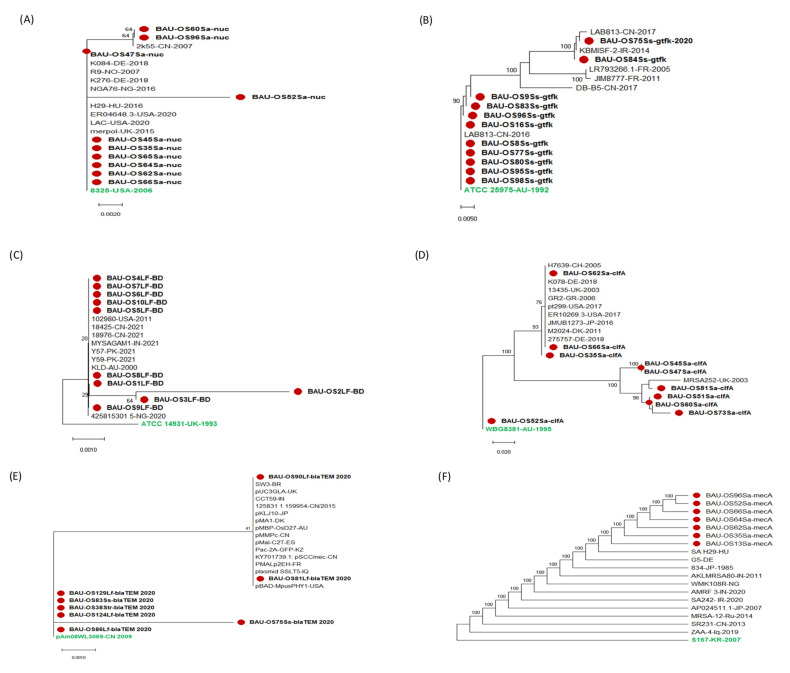
Maximum likelihood evolutionary tree based on nucleotide sequence of different target genes. (**A**) *nuc* gene of *S. aureus* (266 bp). (**B**) *gtfK* gene of *S. sobrinus* (516 bp). (**C**) *lf* gene of *L. fermentum* (317 bp) (**D**). Virulence *clfA* gene of *S. aureus* (943 bp). (**E**) β-lactamase resistance gene *blaTEM* of *Streptococcus* and *Lacotobacillus* (861 bp). (**F**) Methicillin-resistance gene *mecA* of *S. aureus* (608 bp). Bootstrap values (1000 replication) were shown next to the nodes. All ambiguous positions were removed for each sequence pair (pairwise deletion option). Red circles indicated Bangladeshi isolates. The tree is rooted to the green color taxon from a different country.

**Table 1 pathogens-11-00890-t001:** Prevalence of bacteria from the samples originated from oral and periodontal diseases among children in Mymensingh.

Species	No. of Samples		Number of Positive		Total %
	With Signs	Without Signs	With Signs	Without Signs	
*Staph. aureus*	100	31	51	13	48.9
*S. salivarius*	100	31	45	15	45.8
*S. mutans*	100	31	15	7	16.8
*L. fermentum*	100	31	16	2	13.7
*S. sobrinus*	100	31	1	0	0.8

**Table 2 pathogens-11-00890-t002:** Antimicrobial resistance pattern of *S. aureus*, *Streptococcus* spp., and *L. fermentum*.

Name of Antibiotics	No. (%) of Isolates								
	*S. aureus* (n = 64)			*Streptococcus* spp. (n = 83)			*L. fermentum* (n = 18)		
	R	I	S	R	I	S	R	I	S
Clindamycin (CD)	11 (17.2)	8 (12.5)	45 (70.3)	30 (36.1)	-	53 (63.9)	7 (38.9)	1 (5.6)	10 (55.6)
Azithromycin (AZM)	38 (59.4)	7 (10.9)	19 (29.7)	24 (28.9)	18 (21.7)	41 (49.4)	3 (16.7)	3 (16.7)	12 (66.7)
Ciprofloxacin (CIP)	3 (4.7)	14 (21.9)	47 (73.4)	4 (4.8)	1 (1.2)	78 (94.0)	4 (22.2)	1 (5.6)	13 (72.2)
Amoxicillin (AMX)	58 (90.6)	1 (1.6)	5 (7.8)	76 (91.6)	-	7 (8.4)	18 (100)	-	-
Chloramphenicol (C)	-	3 (4.7)	61 (95.3)	8 (9.6)	12 (14.5)	63 (75.9)	1 (5.6)	2 (11.1)	15 (83.3)
Cefuroxime (CXM)	59 (92.2)	1 (1.6)	4 (6.3)	66 (79.5)	-	17 (20.5)	18 (100)	-	-
Erythromycin (E)	39 (60.9)	11 (17.2)	14 (21.9)	35 (42.2)	29 (34.9)	19 (22.9)	6 (33.3)	-	12 (66.7)
Moxifloxacin (MO)	15 (23.4)	3 (4.7)	46 (71.9)	4 (4.8)	3 (3.6)	76 (91.6)	18 (100)	-	-
Metronidazole (MT)	64 (100)	-	-	82 (98.8)	1 (1.2)	-	17 (94.4)	-	1 (5.6)
Tetracycline (TE)	8 (12.5)	6 (9.4)	50 (78.1)	26 (31.3)	15 (18.1)	42 (50.6)	7 (38.9)	2 (11.1)	9 (50)
Cephradine (CH)	61 (95.3)	-	3 (4.7)	81 (97.6)	-	2 (2.4)	16 (88.9)	2 (11.1)	-
Cefixime (CFM)	62 (96.9)	2 (3.1)	-	63 (75.9)	5 (6)	15 (18.1)	18 (100)	-	-

n = number of isolates tested; R = resistant; I = intermediate; S = susceptible; - = not used.

**Table 3 pathogens-11-00890-t003:** Frequency of resistance genes in *S. aureus*, *Streptococcus* spp. and *L. fermentum*.

Target Gene	Organism	Number of AMR Genes Positive Isolates	Percentage (%)
*blaTEM*	*S. aureus* (n = 64)	3	4.7
	*Streptococcus* spp. (n = 83)	10	12.1
	*L. fermentum* (n = 18)	11	61.1
*cfxA*	*S. aureus* (n = 64)	60	93.8
	*Streptococcus* spp. (n = 83)	79	95.2
	*L. fermentum* (n = 18)	13	72.2
*nim*	None	None	None
*mecA*	*S. aureus* (n = 64)	8	12.5

## Data Availability

The sequences analyzed in this study were downloaded from the NCBI (https://www.ncbi.nlm.nih.gov/nucleotide/, accessed on 12 April 2022) database. The sequence metadata and other related documents generated for bioinformatics are available as Appendix A.

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
