# Peer review of "Prevalence of Methicillin and β−Lactamase Resistant Pathogens Associated with Oral and Periodontal Disease of Children in Mymensingh, Bangladesh"

_pathogens, 2022, doi:10.3390/pathogens11080890_

Round 1

Reviewer 1 Report

Manuscript “Prevalence and occurrence of methicillin and β−lactamase resistant pathogens …” by Sultana S. et al. describes the data on isolation and characterization of pathogenic bacteria causing oral and periodontal diseases in children in Bangladesh. 131 isolates were collected at Mymensingh Medical College Hospital and characterized using microbiological, molecular biology and bioinformatic techniques. Bacteria were screened for antimicrobial susceptibility to 12 antibiotics and the presence of six virulence and three antibiotic resistance genes.

The manuscript is well written and detailed. I have only minor questions requiring a response from the authors. Further, the article can be accepted for publication.

 Comments:

1)                 The data in Fig. 2 is not clear in part: for example, all strains of L. fermentum were found to be MDR, but the number of strains resistant to 5-8 and 9-12 antibiotics is rather small. For S. aureus, it is not clear how 73.4 % of strains can be resistant to 5-8 antibiotics and 77.8 % of strains to 9-12 antibiotics. What is the meaning of the division into groups of 1-4, 5-8 and 9-12 antibiotics? Check the labels of the values of the x-axis: apparently it is not necessary to repeat “Number”.  

2)                 Was a negative PCR control with blaTEM primers without template DNA performed? As it is known that a recombinant Taq-polymerase can include a TEM type beta-lactamase gene.

3)                 Why did you use a 100bp DNA ladder for electrophoresis while amplified products were of larger size?

4)                 Since 131 samples were examined and the number of bacteria exceeded this value, did several swab samples contain more than one bacteria?

5)                 The description of 131 samples includes that 100 of them were isolated from patients with clinical symptoms; the rest corresponded to asymptomatic cases. However, the analysis of the results does not include a comparison of these samples. It would be interesting.

6)                 Resolution of text in Fig. 3 should be increased.

7)                 Suppl. Table S1 is missing.

 Minor comments:

Line 25 Replace isolate with isolates

Line 25-26 Please check: antibiotics or antibiotic classes?

Line 29 Beta-lactam resistant, not beta-lactamase

Line 38-39 Ecological community of biofilms is formed by microorganisms mentioned, so the phrase should be reformulated

Line 113 Should be MDR S. aureus....?

Line 131 L. Fermentum

Line 306 Check 1000C, line 309 200C

Line 320 forward and reverse primers

Line 319 Source of the master mix?

Author Response

Reviewer 1

Comments:

  1. The data in Fig. 2 is not clear in part: for example, all strains of L. fermentum were found to be MDR, but the number of strains resistant to 5-8 and 9-12 antibiotics is rather small. For S. aureus, it is not clear how 73.4 % of strains can be resistant to 5-8 antibiotics and 77.8 % of strains to 9-12 antibiotics. What is the meaning of the division into groups of 1-4, 5-8 and 9-12 antibiotics? Check the labels of the values of the x-axis: apparently it is not necessary to repeat “Number”.  

Response: We have worked on Figure 2 and in the text, we have cleared the explanation. The overall prevalence of MDR S. aureus, Streptococcus spp. and L. fermentum were 92.2%, 95.1% and 100%, respectively. Among these three bacteria, the higher proportions of the isolates were resistant to 3–4 antimicrobial classes than 5−6 and ≥ 7 classes of antimicrobials (Figure 2). Among 64 S. aureus, 83 Streptococcus spp. and 18 L. fermentum isolates, all the isolates were resistant to at least one, and up to 12 antimicrobial agents tested (yellow shade, in line 125-129).

  1. Was a negative PCR control with blaTEM primers without template DNA performed? As it is known that a recombinant Taq-polymerase can include a TEM type beta-lactamase gene.

Response: Yes, negative PCR control with blaTEM primers without template DNA was performed. Only Nuclease free water was used in negative controls (yellow shade, line 343)

  1. Why did you use a 100bp DNA ladder for electrophoresis while amplified products were of larger size?

Response: Actually, 100bp DANN ladder ranges from 100 to 1500bp. Therefore, it is valid to use. However, we have changed it to “appropriate” DNA ladder (yellow shade, in line 345-346)

  1. Since 131 samples were examined and the number of bacteria exceeded this value, did several swab samples contain more than one bacteria?

Response: Yes, several swab samples contain multiple coinfection (yellow shade, line 96-97).

  1. The description of 131 samples includes that 100 of them were isolated from patients with clinical symptoms; the rest corresponded to asymptomatic cases. However, the analysis of the results does not include a comparison of these samples. It would be interesting.

Response: We have added new text and a table (Table 1) regarding this context (yellow shade, line 98-102)

  1. Resolution of text in Fig. 3 should be increased.

Response: Resolution of text in Fig. 3 has increased.

  1. Table S1 is missing.

Response: Now included in supplemental materials

Minor comments:

Line 25 Replace isolate with isolates

Response: Corrected in line 25 (yellow shade)

Line 25-26 Please check: antibiotics or antibiotic classes?

Response: Corrected as “antibiotics” in line 26 (yellow shade)

Line 29 Beta-lactam resistant, not beta-lactamase

Response: Corrected in line 31 (yellow shade)

Line 38-39 Ecological community of biofilms is formed by microorganisms mentioned, so the phrase should be reformulated

Response: Rephrased in line 39-40 (yellow shade)

Line 113 Should be MDR S. aureus....?

Response: Corrected (yellow shade and now in line 108

Line 131 L. Fermentum

Response: Corrected (yellow shade, line 116,125, & 143)

Line 306 Check 1000C, line 309 200C

Response: Checked and corrected (yellow shade, line 315, 323,327 & 329)

Line 320 forward and reverse primers

Response: Added (yellow shade, line 341)

Line 319 Source of the master mix?

Response: Source added (yellow shade, line 341)

Reviewer 2 Report

Comments:

The present original study, evaluates the prevalence of methicillin and β−lactamase resistant pathogens associated with oral mucosa and periodontal tissues in children from Mymensingh (Bangladesh).

The study protocol is not completely clear.

Submitted manuscript is well organized and written. Results are clearly presented. Introduction as well as Discussion sections needs to be expanded be slightly expanded.

Editing for English language is needed.

Reviewer’s concerns are detailed below.

Concerns and suggestions:

Title:

·       I would suggest to slightly change the title as follows “the prevalence of methicillin and β−lactamase resistant pathogens associated with oral mucosa and periodontal tissues in children from Mymensingh (Bangladesh)”.

Introduction section:

  • After reference n. 15, please, expand on periodontal microflora (see: Obesity and periodontal disease: a narrative review on current evidence and putative molecular links DOI: 10.2174/1874210601913010526 and Periodontal and peri-implant diseases and systemically administered statins: a systematic review. Dentistry Journal. 2021, 9(9), 100 DOI: 10.3390/dj9090100);

·       I would suggest to remove the period in lines 52-58;

·       Please, consider to re-phrase period in lines 72-75 as follows “However, there is no information on OPD pathogens and also about AMR harmful microorganisms in children with OPD from Bangladesh”.

Materials and Methods section:

  • Please specify “OPD clinical symptoms” (line 81); add a table or synthesize clinical diagnosis;

·       The study protocol is not completely clear; please, explain the study design.

Discussion section:

·       (lines 204-206) To period “Dental patients usually take antibiotics primarily to treat postoperative and secondary 204 infections. β−lactams are one of the most frequently prescribed classes of antibiotics for 205 both medical and dental purposes”, please, add the use of antibiotic administration following tooth extraction to prevent alveolitis (Effectiveness of antibiotics in preventing alveolitis after erupted tooth extraction : a retrospective study. oral dis 2020;26(5):967-973.  DOI: 10.1111/odi.13297).

·       Please, add the clinical translation and relevance of presented findings.

Author Response

Reviewer 2

Concerns and suggestions:

Title:

  1. I would suggest to slightly change the title as follows “the prevalence of methicillin and β−lactamase resistant pathogens associated with oral mucosa and periodontal tissues in children from Mymensingh (Bangladesh)”.

       Response: Thank you for your suggestion. We have deleted the word “occurrence” from the title as suggested.

       Introduction section:

After reference n. 15, please, expand on periodontal microflora (see: Obesity and periodontal disease: a narrative review on current evidence and putative molecular links DOI: 10.2174/1874210601913010526 and Periodontal and peri-implant diseases and systemically administered statins: a systematic review. Dentistry Journal. 2021, 9(9), 100 DOI: 10.3390/dj9090100);

Response: Added an extra sentence in line 52-54 and included two new references [16,17] (yellow shade)

  1. I would suggest to remove the period in lines 52-58;

Response: Done and can be seen in line 59-61 (yellow shade)

  1. Please, consider to re-phrase period in lines 72-75 as follows “However, there is no information on OPD pathogens and also about AMR harmful microorganisms in children with OPD from Bangladesh”.

Response: Edited in line 75-76 (yellow shade)

Materials and Methods section:

  1. Please specify “OPD clinical symptoms” (line 81); add a table or synthesize clinical diagnosis; The study protocol is not completely clear; please, explain the study design.

Response: We have added some text for more clarification. OPD symptoms are described in line 274-276

Discussion section:

  1. (lines 204-206) To period “Dental patients usually take antibiotics primarily to treat postoperative and secondary 204 infections. β−lactams are one of the most frequently prescribed classes of antibiotics for 205 both medical and dental purposes”, please, addthe use of antibiotic administration following tooth extraction to prevent alveolitis (Effectiveness of antibiotics in preventing alveolitis after erupted tooth extraction : a retrospective study. oral dis 2020;26(5):967-973.  DOI: 10.1111/odi.13297).

Response: We have added a sentence in line 228 and included the above suggested reference [51] (yellow shade).

  1. Please, add the clinical translation and relevance of presented findings.

Response: The clinical translation and relevance was added in line 265-267 (yellow shade)

Reviewer 3 Report

Dear Authors,

I have read and analyzed the manuscript “Prevalence and occurrence of methicillin and β−lactamase resistant pathogens associated with oral and periodontal surface of children in Mymensingh, Bangladesh”. The performed research aimed to identify the occurrence of oral and periodontal pathogenic bacteria in young children. I want to congratulate the authors for their work but there are some concerns regarding the performed research and the manuscript:

Major aspects

1.     Title: what does the term “oral and periodontal surface” refers to? The sampling involved oral swabs but no GCF sampling, therefore the term is unclear. 

2.     More importantly, the “oral and periodontal diseases” is also a very unclear term. In my opinion, Materials and Methods section lacks information regarding the found pathologies in the investigated children (carious, endo, perio). Also, a very important aspect, there is no information regarding the treatment history of the subjects, including previous drug regimens.

Minor aspects

1.     The manuscript typing needs to be re-checked. The are some typing errors, lack of spacing between the reference brackets and the previous word or extra-spacing; the “degree” sign is missing in lines 306 and 309.

2.     Bacterial species need to be italicized.

Author Response

Reviewer 3

Major aspects

  1. Title: what does the term “oral and periodontal surface” refers to? The sampling involved oral swabs but no GCF sampling, therefore the term is unclear. 

Response: We have collected only oral swabs so; we modified the word “oral and periodontal surface” to “oral and periodontal disease” in the title and throughout the manuscript.

  1. More importantly, the “oral and periodontal diseases” is also a very unclear term. In my opinion, Materials and Methods section lacks information regarding the found pathologies in the investigated children (carious, endo, perio). Also, a very important aspect, there is no information regarding the treatment history of the subjects, including previous drug regimens.

Response: Now we have described the “oral and periodontal diseases” (yellow shade, in line 274-279). Treatment history is unknown so omitted from manuscript.

Minor aspects

  1. The manuscript typing needs to be re-checked. The are some typing errors, lack of spacing between the reference brackets and the previous words or extra-spacing; the “degree” sign is missing in lines 306 and 309.

Response: We have gone through the manuscript and have considered the spacing, signs and lines (yellow shade)

  1. Bacterial species need to be italicized.

Response: Done

Round 2

Reviewer 2 Report

I congratulate the Authors for the work done

Reviewer 3 Report

The authors have responded to the recommendations.